# The Effects of SRT1720 Treatment on Endothelial Cells Derived from the Lung and Bone Marrow of Young and Aged, Male and Female Mice

**DOI:** 10.3390/ijms222011097

**Published:** 2021-10-14

**Authors:** Ushashi Chand Dadwal, Fazal Ur Rehman Bhatti, Olatundun Dupe Awosanya, Caio de Andrade Staut, Rohit U. Nagaraj, Anthony Joseph Perugini, Nikhil Prasad Tewari, Conner Riley Valuch, Seungyup Sun, Stephen Kyle Mendenhall, Donghui Zhou, Sarah Lyn Mostardo, Rachel Jean Blosser, Jiliang Li, Melissa Ann Kacena

**Affiliations:** 1Department of Orthopaedic Surgery, Indiana University School of Medicine, Indianapolis, IN 46202, USA; udadwal@iupui.edu (U.C.D.); fazal_rehman81@hotmail.com (F.U.R.B.); oawosany@indiana.edu (O.D.A.); cadeand@iu.edu (C.d.A.S.); runagara@iu.edu (R.U.N.); aperugini139@marian.edu (A.J.P.III); ntewari@iu.edu (N.P.T.); sunse@iu.edu (S.S.); skmenden@iupui.edu (S.K.M.); dozhou@iupui.edu (D.Z.); sm89@iu.edu (S.L.M.); rblosser@iu.edu (R.J.B.); 2Richard L. Roudebush VA Medical Center, Indianapolis, IN 46202, USA; 3Department of Biology, Indiana University Purdue University Indianapolis, Indianapolis, IN 46202, USA; crvaluch@iu.edu (C.R.V.); jilili@iu.edu (J.L.)

**Keywords:** angiogenesis, endothelial cells, bone, lungs, aging, Sirtuin 1

## Abstract

Angiogenesis is critical for successful fracture healing. Age-related alterations in endothelial cells (ECs) may cause impaired bone healing. Therefore, examining therapeutic treatments to improve angiogenesis in aging may enhance bone healing. Sirtuin 1 (SIRT1) is highly expressed in ECs and its activation is known to counteract aging. Here, we examined the effects of SRT1720 treatment (SIRT1 activator) on the growth and function of bone marrow and lung ECs (BMECs and LECs, respectively), derived from young (3–4 month) and old (20–24 month) mice. While aging did not alter EC proliferation, treatment with SRT1720 significantly increased proliferation of all LECs. However, SRT1720 only increased proliferation of old female BMECs. Vessel-like tube assays showed similar vessel-like structures between young and old LECs and BMECs from both male and female mice. SRT1720 significantly improved vessel-like structures in all LECs. No age, sex, or treatment differences were found in migration related parameters of LECs. In males, old BMECs had greater migration rates than young BMECs, whereas in females, old BMECs had lower migration rates than young BMECs. Collectively, our data suggest that treatment with SRT1720 appears to enhance the angiogenic potential of LECs irrespective of age or sex. However, its role in BMECs is sex- and age-dependent.

## 1. Introduction

Osteoporosis is the most common musculoskeletal disease that affects 10 million Americans, placing them at an increased risk for developing fractures [1]. This disease burden disproportionately affects the elderly population, with 30% of patients experiencing fractures as the primary injury and contributing to over 50% of all hospital admissions [2]. Elderly patients also experience delayed fracture healing, decreased quality of life, and increased morbidity and mortality [3,4,5,6] due to impaired vasculogenesis and angiogenesis [7], delayed cellular responsiveness, and age-dependent decrease in cellular potential [8]. Animal studies have demonstrated that increased age has negative effects on angiogenesis at a fracture site and might subsequently delay fracture healing or cause non-union of fractures [9,10]. Importantly, different tissue types have blood vessels that have specific function and characteristics [11,12]. Taken together, this suggests that age-related alterations in function or characteristics of vasculature may cause impaired bone healing. However, studies that elucidate the influence of endothelial cells (ECs) and their angiogenic potential are lacking.

Sirtuin 1 (SIRT1), a NAD+-dependent histone deacetylase, is implicated in several biologic processes like angiogenesis, apoptosis, senescence, and proliferation [13,14,15,16,17,18]. SIRT1 is highly expressed in ECs [19] and studies in mice show that aging lowers endothelial expression of SIRT1 while endothelial SIRT1 over-expression causes opposite effects, implying SIRT1 counteracts vascular aging [20,21,22,23]. Moreover, SIRT1 overexpression or pharmacological activation via SRT1720 has been shown to increase bone mass and decrease age-dependent bone loss [21,24,25,26,27,28,29,30,31,32,33]. However, the role of SIRT1 as a potential therapeutic target to both counteract vascular aging in bone while improving bone healing remains to be investigated.

In order to begin to understand the role of SRT1720 treatment on EC function in aging, here, we examined bone marrow and lung ECs (BMECs and LECs, respectively) derived from young (3–4 month) and old (20–24 month) mice.

## 2. Results

For ease of presentation, we will typically first present the LEC data followed by BMEC data. We will present male data followed by female data and we will present vehicle treated (control) data followed by SRT1720 treatment data.

### 2.1. Characterization of Lung and Bone Marrow Endothelial Cells

As expected, mRNA expression of CD31 was observed in all groups (irrespective of treatment, age, and/or sex) in both LECs and BMECs (Appendix A).

### 2.2. Proliferation of Lung and Bone Marrow Endothelial Cells

LECs and BMECs were cultured for 48 h and the number of crystal violet stained cells/field of view were counted. Male and female BMECs were initially treated with a concentration of 0, 0.1, 0.3, or 0.9 μM SRT1720. Appendix A demonstrates that the 0.3 μM SRT1720 dose resulted in changes compared to control whereas, 0.9 μM SRT1720 dose was detrimental to BMECs. Therefore, for subsequent studies, ECs were treated with 0.3 μM SRT1720. LECs exhibited an almost 50% increase in proliferation when treated with SRT1720 irrespective of sex or age (Figure 1A). However, an almost 100% increase in proliferation was observed in the SRT1720 treated old female BMECs when compared to the control old female BMECs (Figure 1B). These data appear to suggest that the proliferative response of ECs to SRT1720 is tissue specific as well as age and sex dependent.

### 2.3. Gene Expression in Endothelial Cells

In LECs and BMECs, mRNA expression of the following angiogenic-related genes was determined: fms-related tyrosine kinase-1 (FLT-1), Angiopoietin 1 (ANGPT-1), and Angiopoietin 2 (ANGPT-2).

FLT-1, a member of the vascular endothelial growth factor receptor (VEGFR) family is thought to promote EC survival and angiogenesis [34,35]. FLT-1 mRNA expression was significantly decreased by more than 50% in control old LECs compared to control young LECs in both males and females (Figure 2A). SRT1720 treatment increased FLT-1 mRNA expression significantly by almost 40% in old male LECs when compared to control old male LECs. In control male BMECs, no differences were observed in FLT-1 expression between young and old male cells (Figure 2B). However, there was a significant decrease of more than 40% in FLT-1 expression in control old female BMECs when compared to control young female BMECs (Figure 2B). SRT1720 treatment of young male BMECs resulted in a significant, greater than 40% increase in FLT-1 expression when compared to control young male BMECs. However, SRT1720 treatment significantly decreased FLT-1 expression by nearly 50% in old male BMECs. For females, an even more robust decrease was observed when young female BMECs were treated with SRT1720 (almost a 90% decrease) compared to control young female BMECs. On the other hand, there was a significant, almost 110% increase in FLT-1 expression in old female BMECs treated with SRT1720 when compared to control old female BMECs.

Angiopoietin (ANGPT-1) is essential for EC survival, vascular branching, and pericyte recruitment [36]. ANGPT-1 mRNA expression in LECs was shown to be significantly decreased by at least 70% in old male and female cells compared to young male and female cells (Figure 2C). SRT1720 treatment stimulated an approximate 50% significant increase in ANGPT-1 expression in only old female LECs. In BMECs, ANGPT-1 expression was significantly increased by more than 70% in control old males than in control young males. Further, SRT1720 treatment did not change ANGPT-1 mRNA expression in young or old male BMECs (Figure 2D). However, in comparison to young female BMECs, ANGPT-1 mRNA expression decreased significantly by more than 40% in old female BMECs. SRT1720 treatment decreased ANGPT-1 mRNA expression by almost 90% in young female BMECs but did not alter ANGPT-1 mRNA expression in old female BMECs.

Angiopoietin (ANGPT-2) is implicated in baseline endothelial function and mRNA expression [37]. ANGPT-2 mRNA expression was shown to be significantly lower by nearly 40% in old LECs in both males and females compared to young LECs. ANGPT-2 mRNA expression in LECs was not significantly affected by SRT1720 treatment (Figure 2E). In BMECs, ANGPT-2 expression did not differ between young and old males (Figure 2F). However, compared to young female BMECs, ANGPT-2 mRNA expression decreased significantly by 70% in old female BMECs. SRT1720 treatment significantly increased ANGPT-2 mRNA expression by 60% in young male BMECs. Of note, a non-significant 46% increase in ANGPT-2 mRNA expression was observed in old male BMECs. In old female BMECs, SRT1720 treatment significantly increased ANGPT-2 mRNA expression by almost 120%. However, in young female BMECs, SRT1720 treatment significantly reduced ANGPT-2 mRNA expression by 85%.

### 2.4. Angiogenic Potential of Endothelial Cells by Vessel-like Tube Formation

Vessel-like tube formation, an indicator of the functional potential of ECs, was evaluated by plating LECs and BMECs on Matrigel to further examine the effects of age, sex, and treatment. Number of nodes, meshes, vessel-like structures, and path length were quantified (Figure 3A–H). LECs demonstrate a striking increase in all vessel-like tube formation related parameters when treated with SRT1720. This increase is nearly 200% in number of nodes and number of meshes (Figure 3A,C), while vessel-like path numbers and length both increased by nearly 400% (Figure 3E,G). SRT1720 treatment significantly activates vessel-like tube formation in LECs irrespective of sex or age. Despite the overall uniform positive response of LECs to SRT1720 treatment, only some positive influences of SRT1720 on BMECs were observed. Specifically, the number of nodes decreased by more than 75% in old female BMECs compared to young female BMECs (Figure 3B). SRT1720 treatment significantly increased the number of nodes by nearly 800% in old female BMECs when compared to control old female BMECs (Figure 3B). The number of meshes (Figure 3D) was significantly increased by 46% in young male BMECs treated with SRT1720 when compared to control young male BMECs. The only differences noted in the vessel-like path numbers (Figure 3F) and lengths (Figure 3H) were significant reductions of at least 40% in SRT1720 treated old male BMECs when compared to SRT1720 treated young male BMECs.

### 2.5. Migration of Endothelial Cells

Migration in ECs was examined using a live cell imaging platform and was quantified by the IncuCyte system. In LECs, no differences were found in the migration ability of ECs between age, sex, or SRT1720 treatment (Figure 4A). Interestingly, irrespective of treatment, the migration potential of old male BMECs was significantly higher when compared to both young control and young SRT1720 treated male BMECs (Figure 4B). For females, old control BMECs had the highest migration potential followed by young control BMECs, young SRT1720 treated BMECs, and old SRT1720 treated BMECs.

## 3. Discussion

Osteoporosis is an aging disease which leads to physical, cellular, and molecular changes that are not fully understood. Several studies have shown that angiogenesis plays a critical role in bone growth, formation, and healing [7,9,10,38], and is impaired during aging [38,39]. Many factors affect this age-mediated decline in angiogenesis, including endothelial function and physiological differences based on isolation from different tissue types [11]. SIRT1, a member of the sirtuin family, plays an integral role in aging cellular metabolism, vascular dysfunction, and disease. Currently, there are 71 clinical trials focusing on the effects of SIRT1 on various disorders, including vascular system injuries, coronary artery disease, aging, and type 2 diabetes [40,41]. Interestingly, El-Haj, et al. observed reduced SIRT1 expression in female osteoporotic patients [33]. Thus, the potential for SIRT1 as a potential therapeutic target to both counteract vascular aging in bone while improving bone regeneration is an exciting possibility.

In this study, we investigated the angiogenic potential of ECs from lungs and bone marrow derived from young and old, male, and female mice, as well as the influence of SRT1720 treatment on that angiogenic potential. We summarize our findings in Table 1 and Table 2.

One of the dysfunctions of ECs caused by aging is a decrease in proliferation. In this study, LECs and BMECs showed a decrease in proliferation in old ECs when compared to young ECs. We further demonstrated that treatment of ECs with SRT1720 resulted in increases in proliferation in all LECs and in old female BMECs (Table 1 and Table 2). There is also a trending increase in proliferation in young female BMECs and to a lesser extent in male BMECs. It is possible that certain cell populations contained within the more heterogenous BMEC cultures did not respond to SRT1720 treatment as robustly as the more homogenous LEC cultures. This may also indicate that a higher sample size would allow for additional significant differences to be detected. Another important observation was that neither LECs nor BMECs were confluent at the time of fixation, so that also should not impact the results. Additionally, to the best of our knowledge, no studies have reported on the effects of SRT1720 treatment on ECs derived from lungs and bone marrow. SIRT1, a known metabolic regulator and important factor in vascular homeostasis, has been shown to have a 37% decrease in protein content in human arteries of elderly patients [20]. SRT1720, is a small drug molecule that activates SIRT1 and has been shown to improve aging-mediated vascular endothelial dysfunction by improving arterial endothelial dependent dilation [21]. Our data demonstrated that treatment with SRT1720 also enhances cellular proliferation of ECs, and although not specifically demonstrated here, may suggest that a downregulation of SIRT1 may be responsible for the decrease in EC proliferation caused by aging.

Our data showed different mRNA expression patterns for several angiogenic genes examined in lung and bone marrow ECs treated with or without SRT1720. Few significant differences were observed in SRT1720 treated LECs. Indeed, for young male and female LECs, no differences in FLT-1, ANGPT-1, or ANGPT-2 expression were observed with SRT1720 treatment. However, in old male LECs, treatment with SRT1720 resulted a significant increase in FLT-1 expression, while in old female LECs, treatment with SRT1720 resulted in a significant increase in ANGPT-1.

With respect to BMECs, SRT1720 treatment significantly increased FLT-1 expression in young male and old female BMECs, but reduced expression in old male and young female BMECs. Somewhat similarly, SRT1720 treatment significantly increased ANGPT-2 expression in young male and old female BMECs and reduced expression in young female BMECs. With regard to ANGPT-1, SRT1720 treatment only significantly altered ANGPT-1 expression in young female BMECs. Interestingly, SRT1720 has been shown to activate angiogenic genes like FLT-1 and ANGPT-2 during vascular calcification in diabetic mouse models [21,42,43]. Several studies have also shown that aberrant Sirtuin-1 signaling, when present in ECs (likely disease state such as diabetes or aging induced), upregulates several pathways that interfere with angiogenesis such as Notch and Wnt pathways [44]. Taken together, our data appear to demonstrate that treatment with SRT1720 involves several angiogenic pathways being activated and influenced by age and sex. Meanwhile, further investigation is required to fully elucidate these molecular mechanisms, it appears complicated, especially with respect to FLT-1 expression. For example, androgens may enhance the effects of SRT1720 on FLT-1 expression in young males. On the other hand, estrogen may antagonize the effects of SRT1720 on FLT-1 expression in young females. Further, increased estrogen to androgen ratios may inhibit the role of SRT1720 in aged male mice, whereas in aged female mice, lower estrogen levels may improve the effects of SRT1720. Overall, future work examining estrogen, androgen, and estrogen to androgen ratios may fine tune our understanding of the effects of SRT1720 on FLT-1 expression in BMECs in vivo. Likewise, it would be of value to consider the use of sex-hormone related models to further dissect mechanisms of action. Of importance, SIRT1 is known to be regulated by estrogen as evidenced by ovariectomy reducing SIRT1 expression whereas estrogen treatment rescued SIRT1 expression [42,43]. Therefore, the regulation of sex hormones through use of ovariectomy and orchiectomy mouse models in conjunction with fracture injuries treated with or without SRT1720 may be valuable.

Compared to LECs, BMECs exhibited significantly more angiogenic mRNA expression differences following treatment with SRT1720. We speculate that this may occur as LECs are isolated as a homogenous CD31+ population whereas BMECs are isolated as a heterogeneous population [45,46,47,48,49] that is comprised of mature and immature EC cells, stalk and tip cells, and perhaps EC-like cells that assist in EC function [11]. Exploring and comparing the gene profiles between these subpopulations of ECs remains to be investigated but is important to better understand how drugs would impact multiple cell types.

Next, we examined the angiogenic potential of LECs and BMECs following SRT1720 treatment using a Matrigel tube formation assay. In general, the vessel-like tube formation data demonstrated that SRT1720 treatment improved angiogenic potential of LECs. For BMECs, treatment with SRT1720 also resulted in increases in a few vessel-like tube formation parameters. However, there were also several parameters in which no differences were observed. These positive (or at least non-deleterious) effects are consistent with previous studies showing the protective effects of SIRT1 against aging mediated dysfunction utilizing in vivo rodent models or primary murine vascular smooth muscle cells from animal aortas [21,43,50]. Again, the less consistent data observed in BMECs may be due to their heterogeneity. Overall, our data have demonstrated positive effects of SRT1720 treatment on the angiogenic potential of both LECs and BMECs and suggest that treatment with SRT1720 may improve angiogenic potential in aging ECs.

Additionally, endothelial function was further examined in terms of migration potential. Surprisingly, no differences were observed between young or old, male, or female LECs treated with or without SRT1720. However, SRT1720 treatment did not appear to impact migration in young or old male BMECs; but inhibited migration in both young and old female BMECs. Our findings appear to be in conflict with the findings of Potente et al., which showed that SIRT1 knockdown blocked murine EC migration whereas over expression of wild type SIRT1 increased migratory activity of ECs [19]. While more work is required to fully understand these conflicting results, it is feasible that the use of different populations of ECs could be responsible for these conflicting findings.

In conclusion, age-related changes in angiogenic potential in both LECs and BMECs have been examined in this study. Treatment of aged LECs with SRT1720 appears to rescue many of the angiogenic parameters investigated here. However, while BMECs demonstrated a more heterogeneous response to SRT1720 treatment, it appears that the overall effect of SRT1720 is beneficial in regard to angiogenic potential. Collectively, these data support the concept that SRT1720 (or similar SIRT activators) could serve as potential therapeutic options to improve angiogenesis and tissue regeneration in the context of aging.

## 4. Materials and Methods

### 4.1. Mouse Model

This work was conducted in compliance with protocols approved by the Indiana University School of Medicine Institutional Animal Care and Use Committee. Young (3–4 months) and old (20–24 months), C57BL/6J male and female mice were generously provided by the National Institute of Aging (NIA, Bethesda, MA, USA).

### 4.2. Isolation of Mouse Primary Endothelial Cells

LECs were isolated under sterile conditions as previously described [51]. Briefly, a biotin rat anti-mouse CD31 antibody (BD Pharmingen™, San Jose, CA, USA) was conjugated with dynabeads (Thermo Fisher Scientific, Waltham, MA, USA). Mice were then sacrificed, and isolated lung tissue was finely minced in small pieces and digested with 225 U/mL collagenase type 2 solution (Worthington Biochemical Corporation, Lakewood, NJ, USA) for one hour at 37 °C and 5% CO_2_. The minced tissue was then filtered through 70 µm mesh strainer and incubated with CD31-conjugated dynabeads for 1 h at 4 °C. CD31+ cells were separated by applying a magnetic field using Dynamag 2 (Thermo Fisher Scientific, Waltham, MA, USA). Isolated CD31+ cells were plated at a density of 3 × 10^5^ cells/mL, in Endothelial Cell Growth Medium 2 (PromoCell, Heidelberg, Germany) supplemented with Growth Medium 2 Supplement Mix (PromoCell, Heidelberg, Germany) and Penicillin-Streptomycin-Glutamine (Thermo Fisher Scientific, Waltham, MA, USA) in collagen I coated 6-well plate (Corning^®^, Corning, NY, USA). LECs were passaged 2 times before experimental use. We have previously published on the characterization of LECs isolated in this fashion [49]. Specifically, we characterized LECs by examining Tie2 expression as evidenced in LECs isolated from mice in which tdTomato is expressed under the control of the Tie2 promoter following activation with tamoxifen and through immunocytochemical staining of CD31 as well as the functional and mRNA expression assays detailed here.

BMECs were isolated from the femurs, tibiae, and humeri as previously described [45]. Briefly, excess muscle and tissue was removed, and each bone was cut at proximal and distal epiphyses, then inserted into a punctured 0.5 mL tube placed into a larger 1.5 mL tube containing 1 mL of cold 10% α-MEM (Gibco, Grand Island, NY, USA) with 10% FBS (Biowest, Riverside, MO, USA). Tubes were centrifuged at 14,000 RPM for 2 min at 4 °C. Resulting cell pellets were resuspended in Complete Endothelial Cell Growth Media (ScienCell, Carlsbad, CA, USA) and plated into 12-well plates coated with 4 μg/mL fibronectin (Thermo Fisher Scientific, Waltham, MA, USA). Bone marrow from one tibia or one femur was plated per well, whereas bone marrow from 2 humeri were plated per well. BMECs remained in this culture for seven days before being used for experiments. We have previously characterized BMECs isolated from this procedure by a number of criteria, including the examination of Tie2 expression (as evidenced in BMECs isolated from mice in which tdTomato is expressed under the control of the Tie2 promoter following activation with tamoxifen), CD31 immunocytochemical staining, and flow cytometric analysis (CD45- Ter119- CD31+ CD105+ cells) [47,48,49].

### 4.3. In Vitro SRT1720 Treatment

BMECs were plated as described below. The cells were treated with vehicle (no drug treatment) or 0.1, 0.3, or 0.9 μM SRT1720 (BioVision Inc., Milpitas, CA, USA).

### 4.4. Proliferation Assay

For proliferation assays, LECs were seeded at a density of 5 × 10^2^ cells/well in a 24 well plate for 48 h in the presence or absence of 0.3 µM SRT1720 (BioVision Inc., Milpitas, CA, USA). Cells were fixed with 10% NBF, and nuclei stained with 1 µg/mL DAPI solution. Micrographs were taken with EVOS^®^ FL Cell Imaging System and DAPI positive cells were manually counted using ImageJ.1.52a software [52].

With regard to BMEC proliferation, BMECs were seeded in a 96-well plate at a density of 5 × 10^3^ cells/well for 48 h in the presence or absence of SRT1720 (0, 0.1, 0.3, or 0.9 µM). Cells were fixed with 5% NBF at room temperature for 20 min, stained with 0.05% crystal violet for 30 min, washed with water, and dried overnight. Micrographs were taken with the EVOS^®^ FL Cell Imaging System and crystal violet stained cells were manually counted using ImageJ.1.52a software.

### 4.5. Gene Expression Analysis

For gene expression analysis, both LECs and BMECs were plated at a density of 1 × 10^6^ cells/well in a 6-well plate. Cells were treated with or without 0.3 μM SRT1720 for 48 h. All cultures were 85–95% confluent when RNA was collected. Briefly, RNA was isolated from the cells using a RNeasy Mini kit (QIAGEN, Hilden, Germany). The Transcriptor First Strand cDNA Synthesis Kit (Roche, Basel, Switzerland) was used to prepare cDNA from 1 μg of total RNA. Quantitative real-time PCR was performed using Power SYBR™ Green PCR Master Mix (Thermo Fisher Scientific, Waltham, MA, USA) on a CFX96 Touch Real-Time PCR Detection System (Bio-Rad, Hercules, CA, USA). The genes analyzed included: CD31 Antigen (CD31), Fms Related Tyrosine Kinase (FLT-1), Angiopoietin 1 (ANGPT-1), and Angiopoietin 2 (ANGPT-2) (Table 3). GAPDH served as an internal vehicle (Table 3). Relative gene expression was calculated using the 2^−ΔΔ*CT*^ method.

### 4.6. Vessel-like Structure Formation Assay

BMEC and LEC vessel-like structure formation was assessed as previously described [53]. In brief, 50 μL/well of Matrigel basement membrane matrix (Corning^®^, Corning, NY, USA) was added to a 96 well plate and incubated at 37 °C for 45 min. At passage 2 for LECs and passage 1 for BMECs, ECs were plated on the Matrigel at a density of 1 × 10^4^ cells/well in the presence or absence of 0.3 μM SRT1720 and incubated at 37 °C and 5% CO_2_. After 6 h or 8 h for LECs or BMECs, respectively, imaging using Nikon TE2000 was carried out. Vessel-like structure formation was quantified by ImageJ.1.52a software. The number of nodes and meshes was measured by Angiogenesis Analyzer, an automated plugin of ImageJ.1.52a. The number of complete vessels and total vessel-like structure length were collected and analyzed using Simple Neurite Tracer, a plugin of ImageJ.1.52b Fiji software by three independent readers which were blinded to group identity.

### 4.7. Wound Migration Assay

In a 96-well plate, ECs were plated at a density of 1 × 10^5^ cells/well and grown for 24 h until 90–100% confluence. In the middle of each well the IncuCyte^®^ WoundMaker (Essen BioScience, Ann Arbor, MI, USA) was used to create a wound. ECs were the incubated in the presence or absence of 0.3 μM SRT1720. ECs were imaged for 50 h by The IncuCyte ZOOM^®^ Live-Cell Analysis System (Essen BioScience, Ann Arbor, MI, USA) at 10× magnification at time 0 and every 2 h. The images were analyzed by IncuCyte™ Scratch Wound Cell Migration Software (Essen BioScience, Ann Arbor, MI, USA).

### 4.8. Statistical Analysis

Statistical analyses were performed using GraphPad Prism (version 8.0.0 for Windows, GraphPad Software, San Diego, CA, USA). One-way ANOVA analyses were followed by Bonferroni’s post-hoc multiple-comparisons unless otherwise stated in figure legends. Student’s *t*-test was performed to determine significant differences between two groups. Data are represented as mean ± standard deviation (SD) and *p* < 0.05 is considered significant. All experiments were performed with technical triplicates with a minimum of three biological replicates.

## Figures and Tables

**Figure 1 ijms-22-11097-f001:**
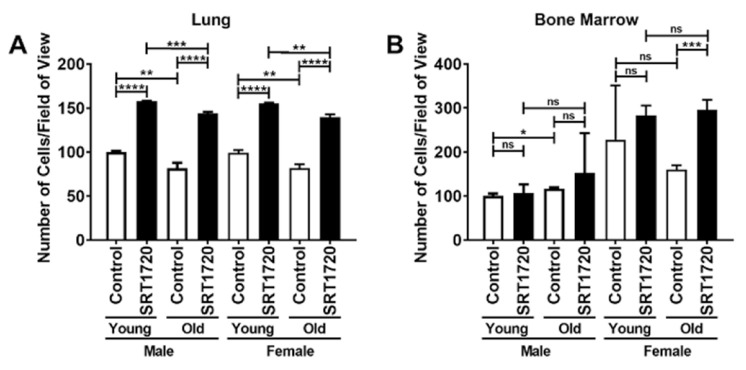
Proliferation of Endothelial Cells (ECs) isolated from the lungs (**A**) or bone marrow (**B**). Data are expressed as the mean ± SD. Young male control parameters were set to 100.0 and parameters in all other samples are shown relative to young male control. *n* = 4 mice/sex/age. ns = not significant, * *p* < 0.05, ** *p* < 0.01, *** *p* < 0.001, and **** *p* < 0.0001.

**Figure 2 ijms-22-11097-f002:**
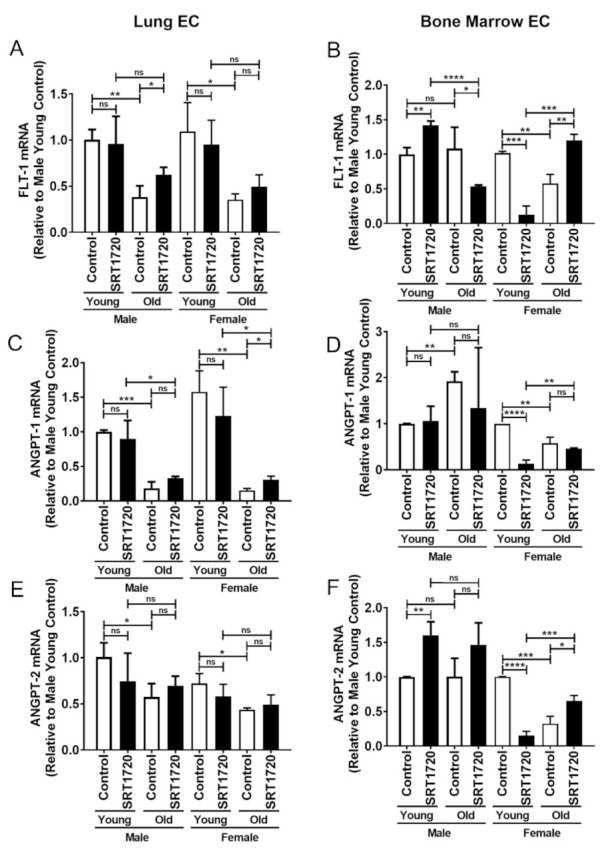
Real-time PCR analysis of ECs isolated from lungs (**A**,**C**,**E**) or bone marrow (**B**,**D**,**F**). Relative mRNA expression was measured for the following genes: fms-related tyrosine kinase-1 (FLT-1) (**A**,**B**), Angiopoietin 1 (ANGPT-1) (**C**,**D**), and Angiopoietin 2 (ANGPT-2) (**E**,**F**). Young male control (vehicle treated) expression was set to 1.0 and expression in all other samples are shown relative to the young male control. Data are expressed as the mean ± SD. An unpaired Student’s *t*-test was used to identify statistically significant differences. *n* = 3 mice/sex/age. ns = not significant, * *p* < 0.05, ** *p* < 0.01, *** *p* < 0.001, and **** *p* < 0.0001.

**Figure 3 ijms-22-11097-f003:**
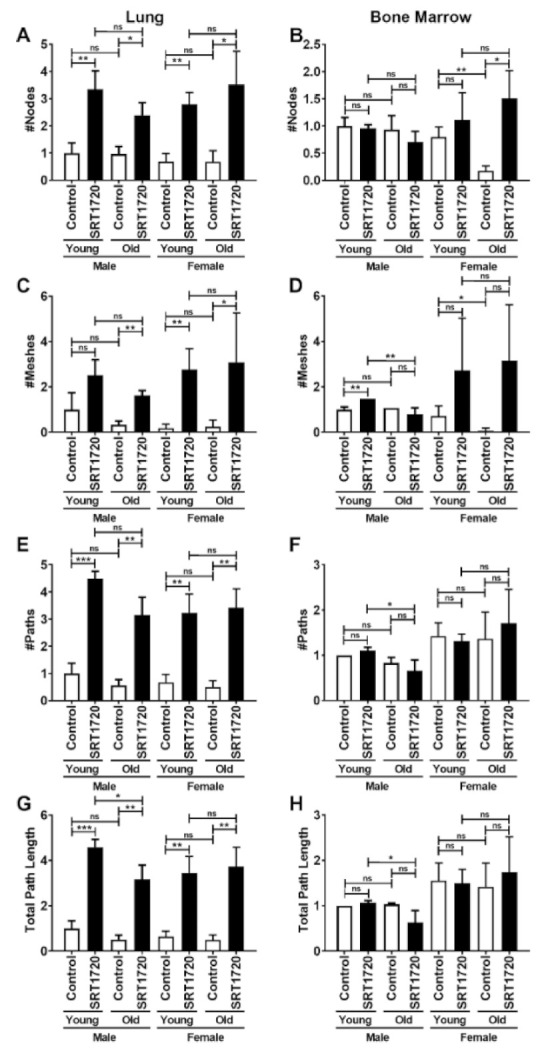
Angiogenesis parameters associated with ECs isolated from lungs (**A**,**C**,**E**,**G**) or bone marrow (**B**,**D**,**F**,**H**). The number of nodes (**A**,**B**), the number of meshes (**C**,**D**), the number of paths (vessel-like tubes) (**E**,**F**), and the total path length (**G**,**H**) were quantified. Young male control (vehicle treated) expression was set to 1.0 and expression in all other samples are shown relative to the young male control. Data are expressed as the mean ± SD. An unpaired Student’s *t*-test was used to identify statistically significant differences. *n* = 5 mice/sex/age. ns = not significant, * *p* < 0.05, ** *p* < 0.01, and *** *p* < 0.001.

**Figure 4 ijms-22-11097-f004:**
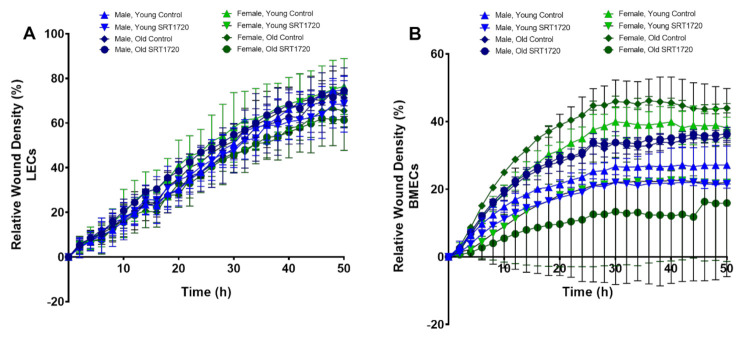
Cell migration of ECs isolated from lungs (**A**) or bone marrow (**B**) as measured by relative wound density (%). The graphs represent data collected for all the groups every 2 h for 50 h. Data are expressed as the mean ± SD. A two-way ANOVA followed by Tukey’s multiple-comparisons test was used to identify statistically significant differences. *n* = 3 mice/sex/age.

**Table 1 ijms-22-11097-t001:** Summary of Lung Endothelial Cell Results.

Outcome Measure		Results	Age				Treatment	
			Male		Female		Control	SRT1720
			*Young*	*Old*	*Young*	*Old*		
Proliferation		▲	X	X	X	X		X
RT PCR	FLT-1	▼		X		X	X	
		▲		X				X
	ANGPT1	▼		X		X	X	
		▲				X		X
	ANGPT2	▼		X		X	X	
Vessel-like Structure Formation	Number of nodes	▲	X	X	X	X		X
	Number of meshes	▲	X	X	X	X		X
	Number of paths	▲	X	X	X	X		X
	total path length	▲	X	X	X	X		X
Migration		⁃⁃⁃						

For the outcome measure, **▲** = Increase, ▼ = Decrease, ⁃⁃⁃ = no change, and “X” in the sex, age, or treatment columns indicate changes in the outcome measure. For example, the first row demonstrates that proliferation was increased in both male and female, young and old lung endothelial cells (LECs) treated with SRT1720 when compared to respective male or female, young LECs.

**Table 2 ijms-22-11097-t002:** Summary of Bone Marrow Endothelial Cell Results.

Outcome Measure		Results	Age				Treatment	
			Male		Female		Control	SRT1720
			*Young*	*Old*	*Young*	*Old*		
Proliferation		▲		X			X	
		▲				X		X
RT PCR	FLT-1	▼				X	X	
		▲	X			X		X
		▼		X	X			X
	ANGPT1	▲		X			X	
		▼				X	X	
		▼			X			X
	ANGPT2	▼				X	X	
		▲	X			X		X
		▼			X			X
Vessel-like Structure Formation	Number of nodes	▼				X	X	
		▲				X		X
	Number of meshes	▼				X	X	
		▲		X				X
	Number of paths	⁃⁃⁃						
	total path length	⁃⁃⁃						
Migration		▲		X			X	X
		▼			X	X		X

For the outcome measure, **▲** = Increase, ▼ = Decrease, ⁃⁃⁃ = no change and “X” in the sex, age, or treatment columns indicate changes in the outcome measure. For example, proliferation was increased in old male controls when compared to the respective young control bone marrow endothelial cells (BMECs).

**Table 3 ijms-22-11097-t003:** Primer Sequences used for Quantitative RT-PCR.

Target	5′-3′ Sequence
GAPDH	**F**: CGTGGGGCTGCCCAGAACAT **R**: TCTCCAGGCGGCACGTCAGA
CD31	**F**: ACGCTGGTGCTCTATGCAAG **R**: TCAGTTGCTGCCCATTCATCA
FLT-1	**F**: CCACCTCTCTATCCGCTGG **R**: ACCAATGTGCTAACCGTCTTATT
ANGPT1	**F**: CACATAGGGTGCAGCAACCA **R**: CGTCGTGTTCTGGAAGAATGA
ANGPT2	**F**: CCTCGACTACGACGACTCAGT **R**: TCTGCACCACATTCTGTTGGA

## Data Availability

The datasets generated during and/or analyzed during the current study are available from the corresponding author on reasonable request.

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
