# Peer review of "The Effects of SRT1720 Treatment on Endothelial Cells Derived from the Lung and Bone Marrow of Young and Aged, Male and Female Mice"

_ijms, 2021, doi:10.3390/ijms222011097_

Round 1

Reviewer 1 Report

Dear colleagues, thank you for this detailed experimental work on the effects of sirtuin activation in angiogenesis. Could you please discuss more in detail what the therapeutical potential of sirtuin activation could mean for the treatment of patients? What is your idea on the sex-different impact of SRT1720 on the FLT-1 expression in BMECs? Do you plan an examination of sex hormone effects additional to SRT1720? Or do you imagine a kind of a hormone-dependent model?

Author Response

We would like to thank the editors and reviewers for the time and effort they have put into improving our manuscript. We have addressed all of the reviewer comments as detailed in our point by point response below. For clarity, we first list the reviewer comments, our responses are in red underneath each comment. We have also made edits to the manuscript and highlighted our edits in red.

Reviewer #1 (Comments to the Author): 

Dear colleagues, thank you for this detailed experimental work on the effects of sirtuin activation in angiogenesis. Could you please discuss more in detail what the therapeutical potential of sirtuin activation could mean for the treatment of patients? Thank you for pointing out an important area for us to expand upon in our manuscript. We have added this to our revised discussion.

What is your idea on the sex-different impact of SRT1720 on the FLT-1 expression in BMECs? This is an important point. While speculative, we have expanded our discussion regarding the sex-based differences of SRT1720 treatment on FLT-1 expression in BMECs and its role in SIRT1 mediated angiogenesis.  

Do you plan an examination of sex hormone effects additional to SRT1720? Or do you imagine a kind of a hormone-dependent model? This is an excellent point for future studies. We have included this in our revised discussion. Specifically, future studies will focus on the important role of sex hormone regulation by using both ovariectomy and orchiectomy mouse models to study SIRT1 activation in the context of aging as well as bone healing and repair (fracture models).   

Reviewer 2 Report

The authors describe the effects of SRT1720 treatment on endothelial cell derived from the lung and bone marrow of young and aged, male and female mice.

Their data suggest that treatment with SRT1720 appears to increase the angiogenic potential of LECs irrespective of age or sex; however, its role in BMECs is sex- and age-dependent.

The publications cited are rich and varied but few are very recent

Their results provide interesting data in relation to the use of SRT1720, but more details need to be added to accept their publication.

When the authors write "minimum of three biological replicates" l.344 we are talking about at least 3 different isolations on 3 different mice?

Figure 2, for example, the number of mice can be specified

After the authors are well aware of the weakness of the BMEC model l.227 and it is true that with these BMECs where the results are less homogeneous and therefore less significant that many questions arise.

This is the first thing you notice when you look at figure 1.

Both for LECs and even more so for BMECs the determination of CD31 gene expression alone is insufficient.

Protein labelling of endothelial markers (microscopy or cytometry) together with a picture of the cell layer would add value to the data.

The purity after isolation could be specified in particular for BMEC.

From the method of isolation of BMECs even if they are plater in a special endothelial cell medium, readers would like to be sure that MSCs are not the majority population.

The number of marrow-derived cells cultured in the 12-well plate should be specified.

For proliferation you seed at a much higher density the BMEC 5000 cells per well in 96 well plate against 500 cells per well in 24 well plate for LEC? is it because they are less proliferating? can the results with SRT1720 be less impressive because the cells are already too confluent?

Can the expression of mRNA ANGPT-1and mRNA ANGPT-2 be evaluated in the same way with regard to the favorable influence of their expression on the angiogenic properties of endothelial cells?

On the basis of which criteria or work the doses of SRT1720 were chosen?

In which plate were the cells seeded at 1x10e6 per well for gene expression analysis? Is there a difference in confluence with the functional assays?

Author Response

We would like to thank the editors and reviewers for the time and effort they have put into improving our manuscript. We have addressed all of the reviewer comments as detailed in our point by point response below. For clarity, we first list the reviewer comments, our responses are in red underneath each comment. We have also made edits to the manuscript and highlighted our edits in red.

Reviewer #2 (Comments to the Author):  

The authors describe the effects of SRT1720 treatment on endothelial cell derived from the lung and bone marrow of young and aged, male, and female mice. Their data suggest that treatment with SRT1720 appears to increase the angiogenic potential of LECs irrespective of age or sex; however, its role in BMECs is sex- and age-dependent. The publications cited are rich and varied but few are very recent. Thank you for pointing out this oversight.  We have now included several additional, more recent references in our revised manuscript.

Their results provide interesting data in relation to the use of SRT1720, but more details need to be added to accept their publication. When the authors write "minimum of three biological replicates" l.344 we are talking about at least 3 different isolations on 3 different mice? Figure 2, for example, the number of mice can be specified.  We appreciate the Reviewer’s attention to detail and apologize for our omissions. We have now added biological replicate details for all reported experimental data. For e.g., line 81 now details biological replicates n = 4 mice/sex/age were used for the proliferation study. Similarly, line 132 now details biological replicates n = 3 mice/sex/age, line 157 now details biological replicates n = 5 mice/sex/age, and line 170-171 details n = 3 mice/sex/age.

After the authors are well aware of the weakness of the BMEC model l.227 and it is true that with these BMECs where the results are less homogeneous and therefore less significant that many questions arise. This is the first thing you notice when you look at figure 1.  We appreciate the reviewer pointing out that the observed results for the BMEC data is variable and heterogenous. This is a common occurrence with bone marrow derived cells as there are several cell types that reside within the bone marrow. It was important for us to highlight this difference, as several studies use cells from a more homogenous tissue i.e. lung endothelial cells or a cell line. This tissue specific difference reflects how therapeutic treatments may differentially impact cells from various tissues. This distinction, in our view is extremely important. We have added some additional verbiage related to this discussion.

Both for LECs and even more so for BMECs the determination of CD31 gene expression alone is insufficient. Protein labelling of endothelial markers (microscopy or cytometry) together with a picture of the cell layer would add value to the data. The purity after isolation could be specified in particular for BMEC. From the method of isolation of BMECs even if they are plater in a special endothelial cell medium, readers would like to be sure that MSCs are not the majority population.  The reviewer brings up an important omission. In our previous publications we have characterized both the LEC and BMEC populations studied here by flow cytometry, immunocytochemistry, and gene expression analysis. In our revised manuscript we reference these important previous characterization studies.  

The number of marrow-derived cells cultured in the 12-well plate should be specified. Unfortunately, we did count the specific number of cells that were plated. Rather, we plated 1 long bone (1 tibia or 1 femur)/well and 2 humeri/well. We have clarified this in our methods.

For proliferation you seed at a much higher density the BMEC 5000 cells per well in 96 well plate against 500 cells per well in 24 well plate for LEC? is it because they are less proliferating? can the results with SRT1720 be less impressive because the cells are already too confluent? The seeding density differences were based upon established protocols from our colleague. As to whether the level of confluence could cause the results with SRT1720 treatment to be less impressive because BMECs are too confluent, we do not believe this to be the case. Indeed, examination of proliferation studies in Figure 1B shows that there is plenty of expansion room for male BMECs (they are the ones that do not show statistically significant differences whereas trending or significant increases are seen in female BMEC cultures). We believe it is more likely that the difference is due to the heterogeneity of BMECs and variability observed in primary cell cultures. We have added related text to our discussion of these results.

Can the expression of mRNA ANGPT-1and mRNA ANGPT-2 be evaluated in the same way with regard to the favorable influence of their expression on the angiogenic properties of endothelial cells? As requested we have tried to ensure that all mRNA ANGPT-1 and ANGPT-2 data were evaluated as was done for FLT-1.

On the basis of which criteria or work the doses of SRT1720 were chosen?  We optimized the dosage of SRT1720 based on proliferation experiments wherein BMECs were treated at three doses – 0.1, 0.3, and 0.9 µM and control. These data are provided in Supplementary Figure 2. Briefly, we observed detrimental effects when cells were cultured with 0.9 µM of SRT1720. While trending differences were observed with 0.1 µM of SRT1720, significant differences were observed in some cultures with 0.3 µM of SRT1720, and thus the choice of concentration.

In which plate were the cells seeded at 1x10e6 per well for gene expression analysis? Is there a difference in confluence with the functional assays? We thank the reviewer for pointing out our omission. The cells were plated in a 6-well plate and there were no differences in confluence. This has been added to the manuscript methods.
